# Extracellular Vesicles in Lung Cancer: Prospects for Diagnostic and Therapeutic Applications

**DOI:** 10.3390/cancers13184604

**Published:** 2021-09-14

**Authors:** Taketo Kato, Jody V. Vykoukal, Johannes F. Fahrmann, Samir Hanash

**Affiliations:** 1Department of Clinical Cancer Prevention, The University of Texas MD Anderson Cancer Center, 1515 Holcombe Boulevard, Houston, TX 77030, USA; tkato1@mdanderson.org (T.K.); jody@mdanderson.org (J.V.V.); jffahrmann@mdanderson.org (J.F.F.); 2The McCombs Institute for the Early Detection and Treatment of Cancer, The University of Texas MD Anderson Cancer Center, Houston, TX 77030, USA

**Keywords:** extracellular vesicles, liquid biopsy, cancer biomarker, tumor-associated antigens, immune therapy, drug delivery

## Abstract

**Simple Summary:**

Extracellular vesicles (EVs) are extensively distributed in various biological fluids, and contain diverse bioactive molecules including proteins, nucleic acids and lipids. They are considered to provide high stability to the associated molecular cargoes because of encapsulation by the lipid bilayer, making them ideal for liquid biopsy and as a drug delivery system. Moreover, EVs can affect immunomodulatory functions, including antigen presentation and immune activation and suppression. Inhibiting the production of tumor-derived EVs can support tumor immunity, and immune cell-derived EVs can be used as an anticancer vaccine. This review summarizes the biological functions and isolation methods of EVs, and explores their diagnostic and therapeutic applications in lung cancer.

**Abstract:**

Extracellular vesicles (EVs) are nano-sized lipid-bound particles containing proteins, nucleic acids and metabolites released by cells. They have been identified in body fluids including blood, saliva, sputum and pleural effusions. In tumors, EVs derived from cancer and immune cells mediate intercellular communication and exchange, and can affect immunomodulatory functions. In the context of lung cancer, emerging evidence implicates EV involvement during various stages of tumor development and progression, including angiogenesis, epithelial to mesenchymal transformation, immune system suppression, metastasis and drug resistance. Additionally, tumor-derived EVs (TDEs) have potential as a liquid biopsy source and as a means of therapeutic targeting, and there is considerable interest in developing clinical applications for EVs in these contexts. In this review, we consider the biogenesis, components, biological functions and isolation methods of EVs, and the implications for their clinical utility for diagnostic and therapeutic applications in lung cancer.

## 1. Introduction

Lung cancer is the worldwide leading cause of cancer-related mortality, resulting in an estimated 1.8 million deaths per year [1]. Because the majority of lung cancers are diagnosed at the advanced stages of disease, early detection is key for reducing mortality. Low-dose CT screening has shown benefits in early detection [2], although it fails to identify some individuals destined to develop advanced-stage lung cancer, and many lung cancers occur in individuals that do not meet current screening eligibility criteria. Thus, there is a need for effective early detection strategies and for more effective lung cancer therapy.

Extracellular vesicles (EVs), including exosomes, transfer a wide variety of biomolecules, including proteins, DNA, RNA and lipids. EVs have been implicated in intercellular communication and are known to modulate tumor–host immune interactions [3]. Extracellular vesicles and their tumor-derived molecular cargoes have recently received considerable interest for liquid biopsy applications in early cancer detection. They are also being applied for the diagnosis of other diseases, including virus and parasitic infections [4,5]. Additionally, there is interest in applying EVs therapeutically by utilizing exosomes as drug delivery systems, by inhibiting EV production or release, or by developing immune therapy applications that target immunomodulatory EV functions. In this review, we focus on the biogenesis, components, and biological functions of EVs in lung cancer. Several EV isolation methods and their possible clinical utility for diagnostic and therapeutic applications are also discussed.

## 2. Biological Role of EVs in Lung Cancer

### 2.1. The Biogenesis of EVs

Extracellular vesicles encompass diverse forms of lipid-bound nanoparticles, ranging from exosomes (30–150 nm) and microvesicles (100–1000 μm) to apoptotic bodies (1–5 μm) (Figure 1A). In addition to varying in size, the biogenesis of these EV types can proceed along differing routes. At the present time, efforts are on-going to identify consensus markers that define each of the various EV populations [6]. To this end, the International Society for Extracellular Vesicles (ISEV) has highlighted the need to clarify and standardize methodology when reporting EV study findings, especially with respect to EV collection, measured physical characteristics, biochemical composition and cell of origin [7].

The activation of cell-specific receptors and signaling pathways that initiate the biogenesis of EVs, in particular exosomes, are highly regulated [8]. These intersect with cell endocytic and intracellular vesicle sorting programs. The first stage of early endosome formation is the fusion of the primary endocytic vesicles [9]. Early endosomes pursue one of two pathways, either returning the cargo to the plasma membrane (recycling endosomes) or by transforming into late endosomes, also known as multivesicular bodies (MVB) (Figure 1A). Intraluminal vesicle (ILV) formation starts in early endosomes with the inward membrane budding rapidly after the recycling of the endosomal cargo to the cell membrane, leading to cargo sequestration and distribution into vesicles [9].

ILV protein sorting is a controlled process, with both endosomal sorting needed for transport (ESCRT)-dependent as well as ESCRT-independent mechanisms having been identified [10]. Four protein members of the ESCRT complex play a vital role in packaging biomolecules inside the lumen of exosomes and in the release of exosomes from the cell [11]. The ESCRT-independent pathway also includes the formation of MVBs and ILVs. Exosome secretion has been shown to require sphingolipid ceramide, with the release of exosomes attenuated after the depletion of neutral sphingomyelinase (nSMase), which plays a crucial role in the sorting of proteolipid proteins into ILVs [12]. Recently, additional molecules, including tetraspanin family proteins [13], lipid raft domains [14], flotillin proteins [15] and HSC70 [16], have been linked with exosome formation pathways. After these steps, MVBs may be forced to fuse and degrade with lysosomes, or they may go into the plasma membrane [17,18] (Figure 1A).

### 2.2. The Roles of EVs in Lung Cancer

Exosomes secreted by lung cancer cells have been found to play important roles in tumor growth, development, invasion, metastasis and immunosuppression, as summarized in Table 1. STAT3-regulated exosomal miR-21, for example, enhances vascular endothelial growth factor (VEGF) expression, and thus stimulates tumor angiogenesis and induces malignant bronchial epithelial cell transformation [19]. Through the NF-ĸB-TLR signaling pathway, lung cancer cell-derived exosomes mediate mesenchymal stem cell transformation towards a pro-inflammatory phenotype, thus promoting the growth of lung cancer in the microenvironment [20]. Exosomes expressing epidermal growth factor receptor (EGFR) expression can stimulate tolerogenic dendritic cells and tumor-specific regulatory T cells (Tregs) that attenuate the role of anti-tumor CD8-positive T cells and promote lung cancer growth [21]. Exosomes derived from metastatic cancer cells and from late-stage lung cancer serum were shown to induce epithelial to mesenchymal transformation (EMT) in human bronchial epithelial cells [22]. Additionally, it has been demonstrated that the exosomes taken up by organ-specific cells contribute to the preparation of the pre-metastatic niche [23]. The proteomic profiling of exosomes revealed distinct patterns of integrin expression in which exosomal integrins α6β4 and α6β1 were found to be associated with lung metastases [24], suggesting exosome-associated integrins could be used to predict organ-specific metastases.

## 3. Clinical Applications of EVs for Diagnosis, Prognosis and Therapy

Liquid biopsy is a minimally invasive method that has potential use for early screening as an alternative or complementary source to tissue biopsy for lung cancer. Exosomes occur in virtually all body fluids, including blood, sputum and pleural effusions, and are considered to provide high stability to the molecular cargoes they convey because of encapsulation by the lipid bilayer [33], making them ideal for liquid biopsies. Exosomes express proteins characteristic of their endosomal origin, including TSG101, ALIX and HSP70 [34], which can be used as markers to differentiate exosomes from other circulating vesicle types. EVs in general are secreted by cells according to a variety of homeostatic and pathologic regulatory programs, and they convey parental cell molecular information that is thought to be more representative of the cell of origin compared to markers released through necrosis or apoptosis [35]. Importantly, EV contents not only mirror the composition of donor cells, but also bear hallmarks of such regulated sorting mechanisms that may also offer diagnostic utility [36]. The various pieces of biomolecular information associated with EVs include proteins, lipids, various metabolites and nucleic acids [37,38] (Figure 1B), offering a diverse array of potential biomarkers for lung cancer risk assessment, early detection, diagnosis, prognostication and surveillance.

### 3.1. EV Isolation Methods in Clinical Practice

Several methods have been described in the literature for isolating EVs from various kinds of samples. These methods optimize various aspects of purity, yield, complexity and cost depending upon their downstream purpose. A widely used standard method is density-gradient ultracentrifugation (DG-UC) [39]. Although DG-UC is straightforward and can isolate relatively pure populations of exosomes, it has some disadvantages, including contamination of lipoprotein particles in the final product, the generation of exosomal aggregates, the duration of the procedure and the price of equipment [40,41,42]. The global survey of EV isolation techniques conducted in 2019 with 600 International Society for Extracellular Vesicles (ISEV) respondents showed that while DG-UC was still the most often used technique, the use of size-exclusion chromatography (SEC) has risen dramatically to more than double its percentage of use in the previous survey from 2015 [43]. The mechanism behind SEC is based on size, and specifically on the hydrodynamic radius of the isolate in question [44]. Although SEC-isolated EVs are relatively pure and the method is efficient, it should be noted that SEC does not enable discrimination between exosomes and microvesicles of the same size [40]. The process of capturing EVs based on immunoaffinity, on the other hand, relies on the isolation of particular EV populations based on surface protein expression, including tetraspanins such as CD9, CD63 and CD81 [45], or EPCAM [46]. While the EV yield is reduced, since only the antibody-recognized EVs are captured, the extracted EVs are of the most narrowly defined purity. Recently, immunoaffinity capture has been applied for clinical EV isolation due to the specificity and simplicity of the approach.

#### 3.1.1. Microfluidics

Microfluidics-based technologies are important techniques for microscale isolation, detection and analysis, and can be applied to EVs and exosomes. These techniques utilize the usual separation determinants such as size, density and immunoaffinity. However, these implementations can accommodate smaller sample volumes to separate EVs of interest [47]. Lee et al. developed an acoustic nanofilter device that continuously separates microvesicles according to size in a contact-free way. The separation uses standing waves of ultrasound to exert an acoustic force on microvesicles and permit discrimination depending on particle size and mass [48]. He et al. developed an integrated microfluidic chip that includes immunomagnetic isolation, chemical lysis and intravesicular protein analysis. The device was also able to assess the total expression and phosphorylation levels of IGF-1R in non-small cell lung cancer (NSCLC) by probing plasma exosomes [49]. Zhang et al. efficiently captured EVs with a microfluidic chip equipped with self-assembled three-dimensional herringbone nanopatterns, which facilitate microscale mass transfer, and increase surface area and sample density to boost exosome binding efficiency and speed. This system detected low levels of tumor-associated exosomes from 2 μL of plasma [50]. These microfluidic systems appear to be ideal for translating liquid biopsies for cancer detection.

#### 3.1.2. Nano-Plasmonic Exosome (nPLEX) Assay

This assay approach is based on the transmission of surface plasmon resonance through periodic nanohole arrays. Plasmonic nanoholes enable the tuning of the probing depth (<200 nm) to balance detection sensitivity to the exosome scale, and the transmitting setup enables the miniaturization of the system as well as the fabrication of a closely packed sensing array. Array elements are functionalized with antibodies to allow the exosomes and proteins found in the exosome lysate to be profiled. A principal advantage of the approach is the achievement of sensitivity 10^4^-fold greater than that of Western blot analysis and 10^2^-fold greater than that of ELISA analyses. Through secondary labeling with additional probes, such as spherical Au nanoparticles, signal amplification can also be achieved on the nPLEX platform [51].

#### 3.1.3. Digital EV Screening Technique (DEST)

The analysis of the relatively scarce circulating EVs released by small, early-stage tumors requires an ultrasensitive diagnostic assay. To enable the detection of such early-stage cancers, Yang et al. developed and optimized a magnetic beads-based digital ELISA assay for EV analyses. The DEST system utilizes magnetic beads coated with capture antibodies for capturing surface or intravesicular proteins from intact or lysed EVs. Biotinylated detection antibodies combined with a tyramide signal amplification step, which catalyzes the addition of biotin groups from tyramide-biotin radicals near HRP, provide improved lower limits of detection [52]. The sensitivity of the approach is 10^4^-fold increased compared to traditional ELISA, which enables the detection of ultra-rare EVs and low abundance proteins [53].

#### 3.1.4. T Cell Immunoglobulin Domain and Mucin Domain-Containing Protein 4 (Tim4)

Tim4 is a type I transmembrane protein expressed in macrophages that recognizes and binds the phosphatidylserine (PS) expressed on exosomes and PS-enriched microvesicles [54]. Nakai et al. used the extracellular region of Tim4 immobilized on magnetic beads to collect small EVs from separate pre-cleared cell-conditioned media and biofluids. Since Tim4 binding to PS is Ca^2+^-dependent, exchange into a buffer containing a chelating agent will readily release intact small EVs. Capturing EVs with Tim4 protein was also applied for the isolation of PS-enriched, large EVs from a 10,000× *g* pellet [55]. Tim4 affinity capture can be applied for the quantification of EVs along with a detection antibody probe in a sandwich ELISA or magnetic beads-based assay format. By employing various antibodies against antigens of interest expressed on EVs, the approach enables high-sensitivity assays using small volumes of sample or biofluid (10–100 µL) [56].

### 3.2. Utility of EVs for Lung Cancer Diagnosis and Prognosis

#### 3.2.1. EV-Associated Proteins

Proteomic profiling has revealed a diversity of EV-associated protein cargoes, including receptors, transcription factors, enzymes, signaling proteins, lipid raft proteins, cytoskeletal and extracellular proteins, vesicle trafficking proteins and immune-interacting proteins [57,58,59]. The surface to volume ratio of EVs is generally three to four orders of magnitude greater than that of the corresponding cells, and there is a concomitant high representation of membrane-associated receptor and cell- and matrix-interacting proteins observed in EVs [60]. Emerging evidence suggests post-translational modifications (PTMs) are one of the important factors mediating the selection of EV-cargoes [61], and alterations in the PTMs of proteins are thought to be a major determinant in the early onset and progression of diseases, including cancer [62]. By coupling a high mannose-type glycan-specific lectin to beads, it was possible to capture small EVs from lung cancer cells [63]. Hoshino et al. investigated the proteomic profiles of extracellular vesicles and particles (EVPs) derived from 426 human samples of various cancer types, and characterized pan-EVP as well as tumor-associated EV protein markers. By using these tissue EV markers or plasma-derived EV cargo, they could distinguish tumors from normal tissues and classify tumors of unknown primary origin. Additionally, they identified Ras Homolog Family Member V (RHOV) to be consistently elevated in lung adenocarcinoma (LUAD) plasma EVs [64]. Sandfeld-Paulsen et al. developed an EV array containing 49 antibodies that could trap and diagnose exosomes in plasma, and discovered that CD151/tetraspanin-24, CD171/L1CAM and tetraspanin-8 were the most important molecules able to distinguish patients with histological lung cancers from cancer-free individuals [65].

#### 3.2.2. EV-Associated RNA

The RNA profile of EVs is distinct from that of cellular RNA, exhibiting a high representation of short RNAs, such as miRNA and mRNA fragments, with little to no intact ribosomal (18S and 28S) RNA [36,66]. Although the mechanism for the sorting of specific mRNA and miRNA into the lumen of EVs and exosomes is not yet fully known, EV-associated RNA has been found to have utility as a diagnostic or prognostic biomarker in lung cancer [67]. It should be noted that patterns of RNA expression differ between various extracellular vesicle and microparticle types, with many of the most abundant miRNAs exhibiting the highest association with non-vesicle compartments [66]. Jin et al. performed miRNA-seq profiling of tumor-derived EVs using plasmas from 46 stage I NSCLC patients and 42 healthy individuals, which demonstrated four miRNA candidates (miR-181-5p, miR-30a-3p, miR-30e-3p and miR-361-5p) for LUAD and three miRNA candidates (miR-10b-5p, miR-15b-5p, miR-320b) for lung squamous cell carcinoma as promising biomarkers for early diagnosis [68]. Wei et al. discovered that gemcitabine-resistant A549 (A549-GR) cells could effectively assemble miR-222-3p into exosomes, which could be transported into parental sensitive cells and promote migration, invasion and gemcitabine resistance [25]. Zhong et al. performed a systematic review of 228 articles related to lung cancer-derived miRNA [69] in clinical cohorts. In this review, 6 of the 228 studies dealt with exosomal miRNA, and miR-10b was an excellent diagnostic exosomal biomarker for NSCLC.

#### 3.2.3. EV-Associated DNA

There are numerous studies indicating the abundant association and specific sorting of circulating cell-free DNA (cfDNA) into EVs. Several recent studies, however, also suggest non-exosomal or even non-vesicular extracellular microparticles as the primary reservoirs of cfDNA [66,70]. If and how DNA species associate with EVs is far from being resolved. The relative fragment size, abundance and precise extracellular entity most closely associated with cfDNA are still being studied, and the biomarker utility of these various compartments remains to be determined [71]. Castellanos-Rizaldos et al. simultaneously captured and interrogated exosomal RNA/DNA and cfDNA to evaluate the T790M detection in NSCLC patient plasma, which achieved 92% sensitivity and 89% specificity [72]. Detection of exosome-based EGFR T790M has demonstrated tremendous promise for the clinical diagnosis in LUAD patients to prevent excessive biopsies of tumors [72]. Abe et al. analyzed the cfDNA of the plasma of lung cancer patients, and found that long fragment cfDNA was detected concomitantly with EVs in the advanced stage; however, short-fragment cfDNA was not in the same EVs fraction. It is conceivable that EVs protect the degradation of long-fragment cfDNA, and can be used as prognostic biomarkers [73].

#### 3.2.4. EV-Associated Lipids

EVs are inherently lipid-bound vesicles, and the biology of the cancer cells they derive from can also be influenced by their lipid content. In particular, enriched sphingolipids, phosphatidylserine and cholesterol bear similarity with lipid raft domains or detergent-resistant membranes [74]. The relative lipid composition of EVs is a function of their biogenesis, and can convey indications of the activation of various stress, metabolic, inflammation or pathologic programs and responses. Fan et al. investigated the lipid profiles of plasma exosomes using ultra high-resolution Fourier transform mass spectrometry for the early detection of NSCLC. By applying Random Forest analysis, 16 lipids of high significance were chosen and their performance assessed, with an area under the receiver operating characteristic curve (AUC) ranging from 0.85 to 0.88 for early- and late-stage cancer versus normal subjects using the selected lipid characteristics [75].

### 3.3. Therapeutic Applications of EVs for Lung Cancer

#### 3.3.1. Immune Cell Inhibitory Effect of EVs

Tumor-derived EVs (TDEs) have emerged as a therapeutic target in lung cancer. In particular, EVs have attracted interest in relation to their expression of tumor-associated antigens (TAAs) and their suppressive effect upon cancer immunity, as well as their support of tumor progression overall [76,77]. The presence of antigen-presenting cells (APCs) and the expression of TAAs, including prenatal exposed antigens, have been shown to contribute to T cell suppression and tumor progression [78]. Lung cancer cell-derived exosomes containing EGFR or other TAAs have been demonstrated to also inhibit the function of CD-positive T cells with anti-tumor functions, thereby promoting the growth of lung cancer and interfering with anti-tumor immunotherapies [21,26]. Exosomal programmed death-ligand 1 (PD-L1) expression induced immune checkpoint activation and has drawn interest as an antigen-independent T cell response. PD-L1 has been found to play a tumor-supporting role in enabling the avoidance of immune surveillance when expressed on the tumor cell surface by interfering with programmed death 1 (PD-1), thereby attenuating the function of cytotoxic CD8+ T cells [27,28].

Lung cancer-derived exosomes inhibit the migration of DCs to lymph nodes by broadly decreasing the C-C/C-X-C chemokine receptors, especially CCR6, CCR7, and CXCR3, on DCs [29]. Various tumor-derived factors have been shown to induce the expansion of myeloid-derived suppressor cells (MDSCs) through multiple pathways, including STAT3 or IL-4Rα-STAT6, resulting in the suppression of T cell function [30]. Investigations of mouse models and human subjects indicate that lung cancer-derived exosomes can activate the immunosuppressive function of MDSCs. Specifically, the heat shock protein 72 (Hsp72) expressed on TDEs has been found to interact with the TLR2 on MDSCs, thus triggering the TLR2/MyD88 signaling pathway, and inducing MDSC function [31].

#### 3.3.2. Suppression of TDEs

TDEs disperse tumor antigens and can serve as key effectors of antitumor immunity. However, as outlined above, TDEs can also act in a decoy role that facilitates tumor resistance to host responses. In addition to the suppression of tumor immunity, TDEs have been found to be involved in inducing lung cancer treatment resistance to radiotherapy, chemotherapy, and targeted therapy [79]. Exosome-mediated resistance to gefitinib in lung cancer cells is reported to be triggered by the miR-21 induction of Akt in lung cancer cells [32]. The inhibition of the development and release of TDEs may therefore offer a novel therapeutic strategy for lung cancer in terms of support for both tumor immunity and the inhibition of drug resistance.

Many pharmacological agents are being investigated to optimally prevent EV release as an anti-cancer therapy. The mechanism of pharmacological inhibition with these agents is based on targeting either (i) EV trafficking (calpeptin, manumycin A and Y27632) or (ii) the metabolism of EV-associated lipids (pantethine, imipramine and GW4869) [80]. Pantethine is a derivative of pantothenic acid (vitamin B5), an intermediate in the production of coenzyme A that plays a role in lipid synthesis and has been used in clinical practice in lowering circulating cholesterol levels. Since cell membrane fluidity is important during the reorganization of the membrane lipid bilayer and the development of microvesicles, pantethine can be applied to impair the shedding of microvesicles [80]. Using MCF-7 cells, the effect of pantethine was studied, and was found to affect EV biogenessis with a cumulative microvesicle reduction of 24% upon pretreatment [81]. More extensive studies of pantethine are warranted to fully understand its potential as a selective inhibitor of lung cancer EVs.

Imipramine is a tricyclic anti-depressant that has gained interest because of its sphingomyelinase acid inhibitory (aSMase) activity. aSMase enzymes catalyze the hydrolysis of sphingomyelin to ceramide [82], which is involved in the development of both exosomes and other EV types as it increases membrane fluidity, and thus facilitates the generation and release of EVs. Studies involving the PC3 cell line have indicated that the biogenesis of microvesicles and exosomes are blocked by imipramine. In this analysis, treatment with 25 μM imipramine resulted in a 77% reduction in the overall release of EVs [83]. Imipramine also induced apoptosis of small-cell lung cancer (SCLC) in both in vitro and in vivo models through neurotransmitter and G protein-coupled receptor (GPCR) signaling proteins [84]. GPCRs and their downstream signaling pathways contribute to the biogenesis, secretion, homing and uptake of EVs [85], consistent with the observation that imipramine inhibits the formation of EVs in SCLC.

An additional approach to decrease the circulating EVs employs cancer-specific EV markers to affinity-capture tumor-secreted vesicles [86]. Marleau et al. initiated clinical trials in cancer patients with a Hemopurifier device, which is operated using a dialysis infrastructure for removing exosomes from circulating blood. This system incorporates a size exclusion and a lectin affinity matrix for capturing nanoparticles with highly expressed mannose glycoprotein surface structures, and was reported to be effective for cleaning 92–99% of exosomes from an input concentration of 10^9^–10^10^ exosomes per mL of plasma [87].

#### 3.3.3. Cancer Vaccine

EVs are known as important mediators of tumor–host immune exchange, and several experiments have shown that EVs from lung tumor cells or immune cells facilitate tumor growth specifically by suppressing anti-tumor immunity. Conversely, modified or engineered EVs can be used as vaccines to provide anti-tumor immunity [88]. Previous experiments have shown that UV-exposed lung cancer TDEs are capable of promoting DC maturation and inducing T cell-dependent anti-tumor immunity, compared to normally secreted TDEs [89,90]. The DNA in TDEs was found to engage the cGAS/STING pathway and possibly be involved in the process underlying DC maturation. Moreover, other research reveals that heat-stressed 3LL lung tumor cell-derived exosomes trigger more successful DC activation than their unprocessed versions, suggesting the content of different inflammatory chemokine ligands in TDEs [91].

Several clinical trials for DC-based vaccines have been performed to target numerous tumor types. Autologous DC multivalent vaccines were investigated in 16 patients with stage IA to IIIB NSCLC who had previously undergone chemotherapy [92]. The apoptotic bodies secreted from irradiated an allogeneic NSCLC cell line overexpressing Her2/neu, CEA, WT1, Mage2 and living cells triggered this DC vaccine. The results from this trial suggest that the vaccine was well accepted, and statistically significant clinical improvement was observed for two of the subjects [93]. Given the importance of immune cell-derived EVs in immune regulation, and specifically of DCs as major immune response coordinators, studies using DC-derived EVs (DEVs) have pioneered the area of therapeutic EV application, and have extended beyond the cancer field [94]. A phase I clinical trial demonstrated the viability of manufacturing autologous DEVs loaded with unique MAGE peptides, as well as the tolerance and safety of the vaccine in MAGE+ NSCLC patients [95]. Based on these trials, a phase II clinical trial with DEVs derived from IFNγ-matured DCs was also initiated [94]. In this study, 22 NSCLC patients were administered four times after chemotherapy with these DEVs loaded with tumor peptides as a maintenance therapy. While no tumor antigen-specific T cell responses could be identified and the primary endpoint of the analysis was not achieved, the procedure did result in progression stabilization in 32% of patients, as well as a long-term stabilization benefit in one patient.

#### 3.3.4. Application of EVs as Drug Delivery Systems

Targeted drug delivery nanocarriers have been demonstrated to be an effective way to treat solid tumors. Nanoparticles such as micelles, polymersomes and liposomes have shown promise, but proven difficult to fully translate into drug delivery agents due to their high clearance rates, toxicity to normal tissues, restricted loading capability and shallow penetration depths [96]. EVs, in contrast, possess inherent stability, biocompatibility, biological boundary permeability and low toxicity, and thus may provide an ideal means to overcome issues associated with other synthetic nanoparticle delivery vectors [97].

Kim et al. developed a process for loading paclitaxel via sonification into exosomes emitted by macrophages. Exosomes modified with the aminoethylanisamide-polyethylene glycol (AA-PEG) vector were studied for targeting the sigma receptor, an often overexpressed NSCLC receptor. In a mouse model of NSCLC, these targeted exosomes were found to improve survival while reducing toxic side effects [98,99]. Srivastava et al. studied the effectiveness of exosomes in encapsulating doxorubicin along with gold nanoparticles (GNPs) as a drug carrier. The efficacy of the exosomes was tested using various NSCLC and lung fibroblast cell lines in an in vitro environment, with the exosome constructs exhibiting preferential cytotoxicity towards the NSCLC cells compared to the control lung fibroblast cells [100]. Agrawal et al. recently explored the oral administration of paclitaxel by loading into milk-derived exosomes (ExoPAC). In a nude mouse model of human lung cancer, the exosome chemotherapeutic showed a greater attenuation of tumor development compared to that seen with paclitaxel administered intravenously [101].

## 4. Conclusions

The development of prognostic, diagnostic and predictive markers for lung cancer is urgently needed, and EVs appear to offer broad potential for applications in minimally invasive liquid biopsies based on recent findings with respect to EV biogenesis, membrane stability and the variety of contained biomolecules. In context, most of the reported findings are based on trials with small numbers of participants, limiting assessments of reproducibility and thus requiring additional multicenter studies to fully test the utility of specific EV-associated markers for screening and diagnostic applications. The ultimate sensitivity and specificity of the myriad of EV-based assays for diagnostic applications in lung cancer still need to be determined and validated.

In terms of therapeutic applications, the use of EVs in antigen-presenting cell systems involving DC-derived exosome is emerging as a powerful technique in lung cancer. Drug delivery systems employing EVs and exosomes also show promise for improving the efficacy and safety of chemotherapy for lung cancer. However, the clinical data are still limited, and further developments and clinical trial tests are needed to fully verify the value of specific EV therapies for lung cancer treatment.

In conclusion, recent findings demonstrate the diverse utility of EVs, and indicate promise for early detection, and the estimation of prognosis or treatment response, in lung cancer. Additionally, EVs appear poised for lung cancer therapeutic applications via their stimulation of immune response, use in anti-tumor vaccination, or capacity to facilitate drug delivery. Further research on EVs in lung cancer will lead to the elucidation of their exact functions and mechanisms of action and will be necessary to ultimately develop optimal approaches for diagnostic and therapeutic applications.

## Figures and Tables

**Figure 1 cancers-13-04604-f001:**
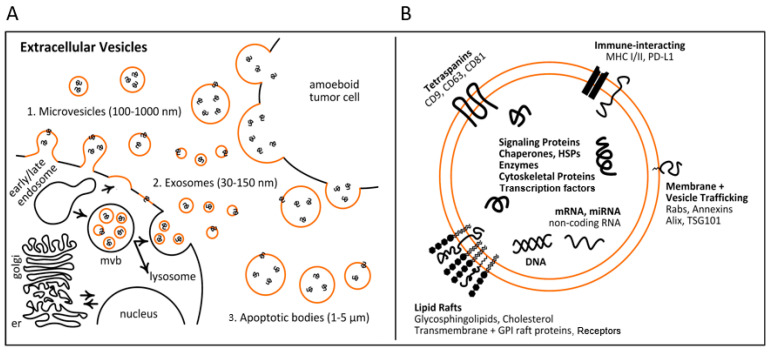
Biogenesis and composition of extracellular vesicles (EVs). (**A**) EVs represent diverse forms of lipid-bound nanoparticles that range from exosomes (30–150 nm) and microvesicles (100–1000 μm) to apoptotic bodies (1–5 μm). EV biogenesis starts with early endosomes formed by the invagination of the plasma membrane. Early endosomes either return the cargo as recycling endosomes to the plasma membrane, or transform into multivesicular bodies (MVBs). MVBs are forced to fuse and degrade with lysosomes, or go into the plasma membrane, and intraluminal vesicles are secreted as exosomes. (**B**) Molecular composition of EVs. EVs contain a variety of cellular components, including nucleic acids, signaling proteins, chaperones, HSP proteins, enzymes and transcription factors as cargoes, and lipid rafts, tetraspanins, immune-interacting molecules and vesicle-trafficking proteins as surface proteins.

**Table 1 cancers-13-04604-t001:** Summary of the biological role of EVs in lung cancer.

Related Molecules in EVs	Influenced Molecules	Function	References
miR-21	VEGF	Enhance the VEGF and angiogenesis	[19]
Exosome-derived lung cancer	NF-ĸB-TLR signaling	Mesenchymal stem cells can be transformed into a pro-inflammatory phenotype	[20]
EGFR		Stimulate tumor-specific regulatory T cells that can limit the role of anti-tumor CD-positive T cells	[21]
Exosome-derived lung cancer		Induce EMT in human bronchial epithelial cells	[22]
Integrin α6β4 and α6β1		Organ-specific lung metastasis	[24]
miR-222-3p		Promote gemcitabine resistance	[25]
EGFR		Inhibit function of CD-positive T cell	[21,26]
PD-L1	PD-1	Avoid immune surveillance and CD8+ T cells	[27,28]
Exosome-derived lung cancer	CCR6, CCR7 and CXCR3	Inhibit migration of DCs	[29]
Exosome-derived lung cancer	STAT3, IL-4Rα-STAT6 pathway	Expand MDSCs and suppress T cell function	[30]
Hsp72	TLR2/My88 signaling pathway	Activate MDSCs	[31]
miR-21	Akt	Gefitinib resistance	[32]

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
