# Peer review of "Extracellular Vesicles in Lung Cancer: Prospects for Diagnostic and Therapeutic Applications"

_cancers, 2021, doi:10.3390/cancers13184604_

Round 1

Reviewer 1 Report

This mini-review gave a concise but clear summary of EV-based applications, in particular related to lung cancer diagnosis and therapy. I see several places the authors can improve the manuscript.

  • 90% of the abstract is about generic description of EVs in terms of their description, biogenesis, functions and applications. They finally mentioned something in the end related to the title, lung cancer. I suggest the authors to modify the abstract to have better summarization of EVs related to lung cancer.
  • The part on EV-related proteins seems quite short, considering the authors are in the field of proteomics. In particular, there are increasing reports on the modified proteins (PTMs) in EVs that are highly relevant to the functions and applications of EVs. They need to expand the session and include PTMs in EVs.
  • The authors have reviewed several methods for the isolation of EVs but did not keep up with recent advances, such as magnetic beads-based isolation and a few others.

Author Response

Comments and Suggestions from reviewer 1  

This mini-review gave a concise but clear summary of EV-based applications, in particular related to lung cancer diagnosis and therapy. I see several places the authors can improve the manuscript.

  • 90% of the abstract is about generic description of EVs in terms of their description, biogenesis, functions and applications. They finally mentioned something in the end related to the title, lung cancer. I suggest the authors to modify the abstract to have better summarization of EVs related to lung cancer.

Thank you for your valuable comment. As we highlight in Table 1, lung cancer derived EVs have been reported to be involved in angiogenesis, EMT, metastasis, drug resistance and immune system suppression. We have revised the Abstract to summarize these functions (P1L26).

  • The part on EV-related proteins seems quite short, considering the authors are in the field of proteomics. In particular, there are increasing reports on the modified proteins (PTMs) in EVs that are highly relevant to the functions and applications of EVs. They need to expand the session and include PTMs in EVs.

Consistent with your suggestion, we added that PTMs are thought to be major determinants in early onset and progression of cancer and that high mannose-type glycan-specific lectin can be used as biomarker for lung cancer (P6L237).

  • The authors have reviewed several methods for the isolation of EVs but did not keep up with recent advances, such as magnetic beads-based isolation and a few others.

Thank you for your suggestion. Magnetic beads-based assay is interesting and important assay to detect small quantities of tumor derived EVs. We added additional information about these approaches in the discussion of Digital EV screening technique (DEST) (P6L207) and T-cell immunoglobulin domain and mucin domain-containing protein 4 (Tim4) (P6L223).

Reviewer 2 Report

The authors provided a well-documented overview of Ev's role and application in Lung cancer. In addition to this, the review could represent a really interesting point of view in a field so dynamic and rich in potential future applications. The field of research focused on exosomes is in continuous evolution and even if the article is well written, the introduction section could be improved with a more general point of view about the application of EVs research in other fields of research adding some recent works related to the importance of exosomes in other diseases (PMID: 31936232). 

I hope that my comments could be useful and I look forward to reading the revised version of the paper.

Good luck.

Author Response

Comments and Suggestions from reviewer 2

The authors provided a well-documented overview of Ev's role and application in Lung cancer. In addition to this, the review could represent a really interesting point of view in a field so dynamic and rich in potential future applications. The field of research focused on exosomes is in continuous evolution and even if the article is well written, the introduction section could be improved with a more general point of view about the application of EVs research in other fields of research adding some recent works related to the importance of exosomes in other diseases (PMID: 31936232). 

Thank you for your affirmative comments. Following your suggestion, we added the application of EVs in other fields especially with respect to virus and parasitic infections in the Introduction (P2L51).